# Biocatalysts Synthesized with Lipase from *Pseudomonas cepacia* on Glycol-Modified ZIF-8: Characterization and Utilization in the Synthesis of Green Biodiesel

**DOI:** 10.3390/molecules27175396

**Published:** 2022-08-24

**Authors:** José Manuel Martínez Gil, Ricardo Vivas Reyes, Marlon José Bastidas Barranco, Liliana Giraldo, Juan Carlos Moreno-Piraján

**Affiliations:** 1Grupo de Investigación Catálisis y Materiales, Facultad de Ciencias Básicas y Aplicadas, Universidad de La Guajira, Rioacha 44001, Colombia; 2Grupo de Investigación Química Cuántica y Teórica, Facultad de Ciencias Exactas y Naturales, Universidad de Cartagena, Cartagena de Indias 130005, Colombia; 3Grupo de Investigación Desarrollo de Estudios y Tecnologías Ambientales del Carbono (DESTACAR), Facultad de Ingenierías, Universidad de La Guajira, Rioacha 44001, Colombia; 4Departamento de Química, Grupo de Investigación en Sólidos Porosos y Calorimetría, Facultad de Cienicas, Universidad de los Andes, Bogotá 111711, Colombia; 5Grupo de Calorimetría, Departamento de Química, Facultad de Ciencias, Universidad Nacional de Colombia, Sede Bogotá, Carrera 30 No. 45-03, Bogotá 111321, Colombia

**Keywords:** lipase, ZIF-8, immobilization, biocatalyst, green biodiesel

## Abstract

This research presents results on the production of biodiesel from the transesterification of acylglycerides present in palm oil, using the biocatalysts ZIF-8-PCL and Gly@ZIF-8-PCL synthesized by immobilization of *Pseudomonas Cepacia Lipase* as catalytic materials and using pure ZIF-8 and Gly@ZIF-8 (modified ZIF-8) as supports. The Gly@ZIF-8 carbonaceous material was prepared by wet impregnation of ZIF-8 with ethylene glycol as the carbon source, and then thermally modified. The calcination conditions were 900 °C for two hours with a heating rate of 7 °C/min in an inert atmosphere. A textural characterization was performed, and results showed superficial changes of materials at the microporous and mesoporous levels for the Gly@ZIF-8 material. Both the starting materials and biocatalysts were characterized by infrared spectroscopy (FTIR) and Raman spectroscopy. During the transesterification, using the two biocatalysts (ZIF-8-PCL and Gly@ZIF-8-PCL), two supernatant liquids were generated which were characterized by infrared spectroscopy (FTIR), gas chromatography coupled to mass spectrometry (GC-MS), and nuclear magnetic resonance (NMR). The results show that the two routes of synthesis of supports from ZIF-8 will be configured as effective methods for the generation of effective biocatalysts for biodiesel production.

## 1. Introduction

The production of biofuels has had a strong increase in recent years, due to the need to improve energy efficiency and the decrease in fossil fuel reserves [1]. In addition, they present low CO_2_ emission rates, which is why they are considered clean and renewable sources of energy, a priority in mitigating the environmental damage caused by the use of fossil fuels [2]. At an industrial level, biodiesel is obtained mainly by transesterification of vegetable oils with methanol or ethanol in the presence of an acid or base catalyst, to produce the corresponding fatty acid methyl or ethyl esters (FAME) and glycerol [3]. Biodiesel is produced mainly by homogeneous acid or base catalysis, with remarkable catalytic activity [4]. However, this process encounters various problems, such as separation, purification, wastewater production, sensitivity to free fatty acids, and recyclability. Homogeneous catalysis is the most widely used technological process for biodiesel production and is based on the use of homogeneous basic catalysts (mainly NaOH or KOH) dissolved in methanol [5]. Although the reactions for obtaining biodiesel by homogeneous catalysis are fast and offer high conversion rates, they present serious drawbacks when the raw material exhibits high acidity [6,7]. In this case, the catalyst must be neutralized and separated from the methyl ester phase at the end of the reaction, with the subsequent generation of a large volume of wastewater [8]. The use of heterogeneous catalysts can help minimize the above problems, including soap production and biodiesel separation and purification [9], and the cost of biodiesel could be drastically reduced. However, the higher the acidity of the feedstock, the lower the conversion efficiency [10]. To produce biodiesel in condensed phase, heterogeneous catalysis proposes the use of catalysts based on zeolites, alkaline earth oxides, carbonates, hydrotalcites, and super acid solids [11,12]. It has been observed that, in the presence of low molecular weight alkalis and alcohols, the transesterification reactions are relatively slow, among other reasons due to the presence of water and free fatty acids, although they are the most used given the accessibility and low cost of alcohol [13,14]. In contrast to base catalysis in acid catalysis, the catalysis can be separated and recycled, separation protocols are reduced, and environmental impact problems are reduced [15]. Various heterogeneous catalysts have been used for the production of biodiesel from different raw materials [16,17,18]. The reuse of the heterogeneous catalyst mainly defines its economic viability. However, current heterogeneous biodiesel technology has not shown any substantial reuse, and therefore heterogeneous catalyst reuse needs to be improved to develop a sustainable biodiesel production technology [19,20]. Many studies have been performed on the synthesis of heterogeneous catalysts to reduce the problems of homogeneous catalysts in biodiesel synthesis. Among them are oxides and derivatives of alkali metals [6,21], alkaline earth metal oxides supported with copper oxide [22], ion exchange resins and sulfated oxides [16,23], heterogeneous catalysts based on waste materials [24], and heterogeneous catalysts based on enzymes [25,26]. In particular, the latter catalysts have awakened great interest in the scientific community [27], especially those resulting from the immobilization of enzymes on MOFs-type supports, because these supports confer greater catalytic activity, greater stability, and allow reuse [28,29] to the new structural configurations in relation to other types of supports. However, although MOFs present a great load capacity due to their large surface areas, their structure tends to be destroyed by their weak coordination interactions [30], so it is necessary to modify their structure by impregnation, humidity, and subsequent calcination, which leads to modification of their textural properties, for example, a reduction of the surface area and an increase in pore size [31]. This modification of the textural properties allows the good anchoring of the enzyme in the support. The lipases of *Pseudomonas Cepacia* have a highly open configuration [32,33], and therefore the substrate with which it interacts must have the sites with the appropriate conformation that facilitates its anchoring. The lipase of *Pseudomonas Cepacia* is characterized by presenting thermal resistance and tolerance to various substances, especially alcohols [34]. If we add to these properties a good anchoring of this enzyme, its thermal resistance would be increased, and therefore the resulting biocatalysts can be used in transesterification reactions that exceed the thermal limits of free lipase [35]. There have been many efforts to improve the problems associated with the use of free lipases in transesterification reactions [36]; however, immobilization is the best mechanism to overcome the drawbacks associated mainly with the deactivation of lipases [37]. The immobilization of lipases, in addition to not generating by-products [38] in the process of converting acylglycerides to biodiesel components, has the additional benefit that the catalyst can be recovered [39]. Several works have shown that the supports produced by the modification of the modified ZIF-8 constitute excellent templates [40] for the anchoring of enzymes, especially lipases from *Pseudomonas Cepacia* [41,42,43,44,45,46]. The success of the support interaction derived from ZIF-8 and lipase lies in the fact that the structure of ZIF-8 can assume new configurations in the presence of molecules with a high electronic density [47]. For this particular case, for a first synthesis route of supports for immobilizing *Pseudomonas Cepacia Lipase*, ethylene glycol was used as a ZIF-8 structure-expanding agent. In this work, ethylene glycol was used as a modifier of the ZIF-8 structure, because it not only stabilizes the MOF structure, but also improves the interaction between the enzyme (*Pseudomonas Cepacia Lipase)* and the substrate (ZIF-8) due to the increase in the pore size of the support and therefore greater anchorage of the *Pseudomonas Cepacia Lipase*. Additionally, it has been demonstrated that the thermal treatment of organometallic MOF structures leads to the formation of derivatives that inherit and amplify the structural properties and morphologies of this type of material, as is the case of the synthesis of Co_3_O_4_ [48]. MOFs have also been used as self-templates for the generation of metal oxides [49], and Pd-loaded ZnO nanotubes have been synthesized from ZIF-8 derivatives [50]. In this context, pyrolysis at 700 and 900 °C of ZIF-67 in the presence of Co(OH) led to the formation of a carbonaceous matrix with a ZIF-67-type structure [51]. In this order of ideas, this work synthesized and characterized biocatalysts with applicability to the conversion, in condensed phase, of triacylglycerides, diacylglycerides, monoacylglycerides present in commercial palm oils, and glycerol as a by-product, in biodiesel components, from two synthesis routes. The first route used the ZIF-8 impregnated with ethylene glycol as support and the second used the Gly@ZIF-8 derived from the thermal calcination at 900 °C of the ZIF-8 modified with ethylene glycol to provide an alternative solution to the problems presented by the traditional catalysts used in the synthesis of biodiesel, namely, through the immobilization of lipases from the species *Pseudomonas Cepacia Lipase* supported on ZIF-8 frameworks. In this research work, two biodiesel synthesis routes are proposed, as alternative processes to overcome the drawbacks presented by the heterogeneous catalysts used in both transesterification reactions and esterification reactions. Likewise, the synthesis of new biocatalysts (ZIF-8-PCL and Gly@ZIF-8-PCL) is described, which were effective to produce fatty acid methyl esters (FAMEs).

## 2. Results

### 2.1. Textural Analysis of Gly@ZIF-8

Figure 1A shows the adsorption–desorption isotherms of N_2_ at 77 K and the pore size distribution curves of Gly@ZIF-8. This behavior indicates that a small number of micropores are found on the surface of Gly@ZIF-8 and the occurrence of mesopores is much higher. This analysis is corroborated with the pore size distribution (Figure 1B). The nitrogen adsorption isotherm at 77 K shows clear evidence of the effect of ZIF-8 modification with ethylene glycol (only the Gly@ZIF-8 isotherm is shown here, because the ZIF-8 isotherm is widely reported). Although the volume of adsorbed nitrogen increases rapidly for the sample at relative pressures below 0.1, the sample has a large adsorption capacity for nitrogen. This is reflected in the evaluated textural parameters, which are shown later.

The nitrogen adsorption isotherm for the Gly@ZIF-8 sample presents a slightly narrower knee at low relative pressures, which probably anticipates a narrow micropore size distribution, and then the presence of the slight development of mesoporosity is observed (a hysteresis loop can be seen above P/P^o^ ~ 0.5). The behavior of the isotherm after adsorbing N_2_ in the first stage is P/P^o^ < 0.1; then, the isotherm reaches a plateau from a relative pressure of 0.1–0.2 and later at pressures higher than 0.85, and then presents a slight ascending curve, which shows the beginning of the formation of the multilayers during the adsorption of the N_2_ molecules, thus approaching the limit value, which is related to the total volume of accessible pores. The sample presents a slight hysteresis loop at a relative pressure between 0.45 and 0.85. This loop is associated with the development of mesoporosity. As mentioned in the first paragraph of this analysis, the shape of the isotherm is observed, not only the isotherm of the Gly@ZIF-8 sample, which has a Type l-b isotherm shape at low relative pressures (below 0.1), which suggests that there is a filling of the micropores (and narrow mesopores). This is confirmed by looking at Figure 2, which shows in detail the low-pressure region when plotting the nitrogen isotherms using a logarithmic scale for this range of relative pressures.

As the relative pressure increases beyond 0.1, the adsorbed volume continues to increase smoothly and continuously, suggesting a large mesopore distribution. A strong increase in the adsorbed volume above 0.75 presents the sample synthesized in this investigation, which may be related to the complex mechanisms that were generated during the modification of ZIF-8, which may also be generating mesoporosity in addition to the development of micropores, as shown by the nitrogen adsorption isotherms in their linear representation. The hysteresis loop according to the IUPAC [52] of the sample corresponds to one type, H3, which indicated that the limit of the desorption branch is normally found in the P/P^o^ induced by cavitation. In summary, ZIF-8 is fundamentally constituted by micropores [53,54], but when this material is subjected to calcination, the new material presents a structural change of the pores and mesoporosity is generated, that is, larger pores. Its presence in the isotherm can be evidenced by the presence of the hysteresis loop. The generation of mesopores is due to the collapse of part of the micropores and the vapors that are generated inside the structure that, when leaving, generate mesoporosity due to the increase in temperature. The textural analysis presented the following results for the Gly@ZIF-8 sample, calculated from the N_2_ isotherm at 77 K: V_m_ (0.161 nm, micropore volume), A_BET_ N_2_ (702 m^2^g^−1^), pore volume (0.306 cm^3^g^−1^), specific surface area (720 m^2^g^−1^), and pore diameter (1.125 nm).

The supports, biocatalysts, and biodiesel were synthesized using a liquid–solid system. The first products of this process, that is, the supports, were characterized in a gas–solid system to determine and compare the surface area and pore size, an analysis of great importance to verify the change in textural properties. This change is necessary to increase the pore size, which will allow or facilitate the anchoring of *Pseudomonas Cepacia Lipase*.

Table 1 presents a summary of the textural characteristics of the catalysts used in this research, including Gly@ZIF-8-PCL.

### 2.2. Fourier Transform Infrared Spectrophotometry (FTIR)

#### Starting Material (ZIF-8) and Biocatalysts ZIF-8-CPL and Gly@ZIF-8-CPL

Figure 3 shows the FTIR spectrum obtained for the ZIF-8 sample. The spectrum presents a band at 420 cm^−1^, which is assigned to the Zn-N bond stretching vibrations corresponding to ZIF-8 [55]. The corresponding bands between 1145 and 1423 cm^−1^ correspond to the bending stretches of the C-N single and C=N double bonds of the imidazolate ring [56]. The bands at 694 and 759 cm^−1^ are assigned to bending vibrations of the C-H single bond. The bands at 952 and 995 cm^−1^ are assigned to the stretching vibrations of the single bond of CO with Zn.

The band at 1585 cm^−1^ is assigned to the stretching vibrations of the C=N bond. When comparing the spectra of ZIF-8 with ZIF-8-PCL (Figure 4A), it was observed that the bands (highlighted in yellow) at 759, 1145, and 1423 cm^−1^ are present in ZIF-8, and these bands are conserved. Additionally, the bands (highlighted in brown) at 420, 694, 995, and 1180 cm^−1^ are present in ZIF-8-PCL but slightly shifted to a lower wavenumber, which corroborates the conservation of the organ structure. The bands (highlighted in blue) at 1739, 2854, and 2924 cm^−1^ are not exclusive to ZIF-8-PCL and are attributed to the interaction of the hydrophobic amino acids of the *Pseudomonas Cepacia Lipase* with the imidazolate rings articulated to Zn in the ZIF-8 structure and polyethylene glycol. The presence of these three bands in Gly@ZIF-8-PCL (Figure 4B) indicates the presence and interaction of *Pseudomonas Cepacia Lipase* with a porous carbon structure that was formed when ZIF-8 was thermally modified in the presence of ethylene glycol as a source of carbon.

### 2.3. Raman Spectroscopy: ZIF-8, Gly@ZIF-8, ZIF-8-CPL, and Gly@ZIF-8-CPL

Figure 5A shows the Raman spectrum of ZIF-8, where it can be seen that the bands with the highest intensity are 101, 177, 1144, 1456, and 2923 cm^−1^. The peak at 177 cm^−1^ is attributed to Zn-N bond vibrations, the peak at 682 cm^−1^ is attributed to imidazolate ring puckering vibrations, the peak at 1144 cm^−1^ is attributed to Zn-N bond vibrations, the peak at 1184 cm^−1^ is attributed to vibrations of the C-N bond, the peak at 1456 cm^−1^ is attributed to bending vibrations of the C-H bond of the methyl group, the peak at 1504 cm^−1^ is attributed to the C-C bond vibrations, and the peak at 2923 cm^−1^ is attributed to the C-H vibrations of the methyl group. When comparing the ZIF-8 spectrum with the Gly@ZIF-8 spectrum (Figure 5B), it can be seen that the new structure does not preserve any ZIF-8 band, but acquires the 1358 and 1593.96 cm^−1^ bands, which are the characteristic signals of this new material, and are attributed to the vibrations of the new C-C and C-Zn bonds in the porous carbonaceous network resulting from thermal modification with ethylene glycol at 900 °C. The spectrum of ZIF-8-PCL (Figure 5C) shows that this new material retains peaks at 370, 682, 1023, 1144, 1184, and 1381 cm^−1^ of ZIF-8 as a starting material.

Likewise, it is observed that the peaks are preserved: 732, 1272, and 1305 cm^−1^, with a slight shift towards a low wavenumber, and the peaks at 839 and 1507 cm^−1^, with a slight shift towards a higher wave number. The other vibration bands are attributed to the interactions of the hydrophobic amino acids of the *Pseudomonas Cepacia Lipase* with the imidazolate ring. When comparing the spectra of GlyZIF-8 with Gly@ZIF-8-PCL (Figure 5D), it is observed that the peaks at 1358 and 1593 cm^−1^ present in GlyZIF-8 are conserved, with a slight shift towards the number of lower waves in Gly@ZIF-8-PCL. The other bands, although of very low intensity, shown in the spectrum of Gly@ZIF-8-PCL are attributed to the interactions of the hydrophobic amino acids with the C-N and C-Zn bonds of the carbonaceous network resulting from the calcination of ZIF-8.

### 2.4. Characteristics of Acidity and Basicity of the Catalyst

To investigate the acid–base characteristics of the materials used and synthesized in this work (ZIF-8, Gly@ZIF-8, and Gly@ZIF-8-PCL) starting from the hierarchical structure of ZIF-8, adsorption of NH_3_ and CO_2_ from these materials has been carried out. Figure 5E,F show the NH_3_ and CO_2_ adsorption profiles for each material. Desorption peaks due to acid sites and basic sites are observed. Table 1 shows the basic quantities corresponding to those determined by the desorption of NH_3_ and the acid quantities corresponding to the sites determined by the adsorption of CO_2_.

As shown in Figure 5E, the desorption peak of ZIF-8 mainly exists in the following temperature ranges: 100–180 °C and 200–300 °C, which correspond to weak acid sites and moderately strong acid sites, respectively. After impregnation with ethylene glycol (Gly), the spectrum changed. Impregnating ZIF-8 with Gly (and obtaining Gly@ZIF-8) in-creased the area of the corresponding catalyst desorption peak at 200–400 °C, indicating that the amount of medium-strong acid increased. This is explained if one considers that ethylene glycol is an acidic compound (pK_a_ = 15.1 ± 0.1), and this is in good agreement with the results shown by the TPD of NH_3_. Finally, when impregnating the previous catalyst with *Pseudomonas Cepacia Lipase* (Gly@ZIF-8-PCL), due to the composition and character of its pH together with the Gly@ZIF-8 compound, it is observed that the peak increases, but only slightly with respect to the previous one (Gly@ZIF-8), showing increased acidity.

The corresponding TPDs with CO_2_ are shown in Figure 5F. The desorption peak area of the Gly@ZIF-8-PCL catalyst at 200–400 °C increased relative to ZIF-8, indicating that the mean strong base amount increased with ethylene glycol treatment. Similar results were obtained by quantifying the basic groups for Gly@ZIF-8-PCL, where with this technique it was determined which acid groups are present. These results clearly show that the compounds prepared from the ZIF-8 hierarchy possess both acid and base sites, and the amount of acid and base increased changes with the impregnation of ethylene-glycol *Pseudomonas Cepacia Lipase*. The acid sites are derived from the Zn, the imidazole groups, specifically from the -N attached to these groups corresponding to ZIF-8 and to the acidity of ethylene glycol. The basicity is lower and comes from the amino groups of the lipase.

In summary, there are a series of catalysts with different textural characteristics and different acid and basic characteristics that will allow them to be tested in the transesterification reaction.

### 2.5. FTIR Biodiesel: Commercial Biodiesel (BIOC) and Biodiesel by Biocatalysis (BIOZIF-8-PCL and BIOGly@ZIF-8-CPL

Figure 6 shows the FTIR spectrum of commercial biodiesel (BIOC), and two types of characteristic vibrational bands for this biofuel can be observed: the first series is made up of the peaks (in yellow) at 1168, 1377, and 2854 cm^−1^, and the second series is made up of the peaks (highlighted in blue) at 1458, 1747, and 2920 cm^−1^. The band at 1458 cm^−1^ falls in the range of the typical peak of the (O-CH_3_) group characteristic of methyl esters, and the bands at 1747 and 1168 cm^−1^ correspond to the carbonyl ester group (-COC) [57]. The peaks at 2920 and 2951 cm^−1^ indicate the presence of moisture [58].

The bands highlighted in yellow in the spectrum of BIOC are conserved without any displacement in BIOZIF-8-PCL and BIOGly@ZIF-8-PCL (Figure 7A,B) (biodiesel synthesized using biocatalysts ZIF-8-PCL and Gly@ZIF-8-PCL). Among the bands highlighted in blue, we find that the 1458 and 2920 cm^−1^ peaks are conserved in the synthesized biodiesels but with a slight shift towards higher wavenumbers, while the 1747 cm^−1^ peak is conserved but with a slight shift to a lower wavenumber. From this behavior we can infer that the liquids synthesized and named BIOZIF-8-PCL and BIOGly@ZIF-8-PCL are true biodiesel.

### 2.6. Gas Chromatography Coupled to Mass Spectrometry

#### 2.6.1. Chromatographic Identification

Figure 8A,B show the chromatograms taken from the different biodiesel samples synthesized using ZIF-8-PCL and Gly@ZIF-8-PCL. The chromatograms show that these biofuels are made up of a mixture of fatty acid esters (FAME). The FAMES that constitute said mixture are the following: methyl palmitate, oleate, and linoleate, with methyl oleate being the FAME with the highest percentages: 36.34% and 36.15%, for ZIF-8-PCL and Gly@ZIF-8-PCL, respectively.

Figure 8A,B show the chromatograms for the biodiesel samples, showing the peaks and retention times run for 40 mins. The peaks eluted according to the number of carbon atoms, first the short-chain peaks, followed by the long-chain peaks, according to the retention time of the FAME^®^ standard presented in Table 2.

#### 2.6.2. Mass Analysis

For the mass analysis of BIOZIF-8-PCL and BIOZIF-8-PCL biodiesel, the database SOFTWARE Xcalibur was used. The identification of the fatty acids present in the synthesized biodiesel is summarized in Table 2, in which only the area percentages and molecular formulas of the fatty acid methyl esters (FAME) present in the synthesized biodiesel are described.

### 2.7. Nuclear Magnetic Resonance (NMR)

In the 1H NMR spectrum (Figure 9A) of the BIOC commercial palm biodiesel, the following signals: 7.3, 5.3, 3.6, 2.1, 1.3, and 0.97 ppm, were observed. These signals are characteristic for this biofuel and therefore are the determinants to establish whether the liquids obtained by the transesterification of commercial palm oil using the biocatalysts BIOZIF-8-PCL and BIOGly@ZIF-8-PCL produced pure biodiesel or biodiesel blends. The presence of these signals in BIOZIF-8-PCL and BIOGly@ZIF-8-PCL is an indicator of whether complete or incomplete transesterification has taken place. As the 1H NMR spectra (Figure 9B,C) of BIOZIF-8-PCL and BIOGly@ZIF-8-PCL present the same characteristic signals of the BIOC spectrum, it can be inferred that the conversion of the acylglycerides present in the oil from palm to biodiesel components was carried out by complete transesterification (see Figure 9D).

Comparing the IR spectra of commercial biodiesel with palm oil (see Figure 9A,B) shows that the significant difference between these two biofuels is the signal at 3.6 ppm.

## 3. Discussion

As the results describe, in this research, green biodiesel was synthesized from the transesterification of acyglycerides present in palm oil, using two new materials: ZIF-8-PCL and Gly@ZIF-8-PCL, as catalysts, which were synthesized by immobilizing the *Pseudomonas Cepacia Lipase* on supports based on a ZIF-8-type structure. The first biocatalyst was obtained by immobilizing *Pseudomonas Cepacia Lipase* in unmodified ZIF-8, and the second biocatalyst was obtained by immobilizing *Pseudomonas Cepacia Lipase* in a new material, Gly@ZIF-8, which was obtained by thermally modifying the ZIF-8 in the presence of ethylene glycol as a carbon source. The thermal modification of the ZIF-8 microporous material [53,54] was evidenced by observing the N_2_ adsorption–desorption isotherms (Figure 1) of Gly@ZIF-8. The textural analysis showed superficial changes from microporous to mesoporous materials in the Gly@ZIF-8; therefore, the calcination of the ZIF-8 material caused a structural change, leading to the new Gly@ZIF-8 material. In particular, the increase in temperature generated significant changes in the textural properties of the carbonaceous material [59]. This structural change begins with the formation of the polyethylene glycol in the presence of ZIF-8, a process that involves a long time [30]. In this work, the time involved was 13 days, corresponding to the maximum turbidity detected, which was assumed as the maximum growth of the polymer based on ethylene glycol. The formed polymer begins to assemble spontaneously [60], which allows its growth from the aqueous ethylene glycol solution until equilibrium is reached. The ZIF-8 is trapped in the formed polymeric network and becomes part of this network, due to the formation of hydrogen bridges between the imidazolate groups present in the ZIF-8 with the hydrogens of the hydroxy groups of the polyethylene glycol that are available, thus forming a new intermediate structure between the ZIF-8 and the ZIF-8-PLC (ZIF-8-PGly) (see Figure 10). These structures present rows or layers of polyethylene glycol, which gives this new structure a large number of hydroxyl groups in layers, which have great affinity with the hydrophobic amino acids present in the *Pseudomonas Cepacia* enzyme [61], allowing a better distribution and anchoring of the lipase, and hence the FAMES components of the biodiesel synthesized from the ZIF-8-PCL biocatalyst have a higher mass percentage (see Table 2). 

As for the Gly@ZIF-8 formation and the calcination of ZIF-8-PGly, although gradual heating was applied, the formation of the carbonaceous network from the thermal decomposition of polyethylene glycol did not protect the ZIF-8 structure, and therefore it suffered textural and structural changes, because the thermal stability of ZIF-8 is around 550 °C [62] and the structural configuration of the ZIF-8-PGly intermediate did not give it the thermal stability to withstand 900 °C. However, the zinc atoms remained in the new structure, forming C-Zn bonds (see Figure 11), a fact that was evidenced by the presence of the 1358 and 1593 cm^−1^ bands in the Raman spectrum of Gly@ZIF-8 (see Figure 5B). 

The presence of Zn atoms in the structure of Gly@ZIF-8 allows lipase anchoring because Zn induces lipase aggregation [63]; however, the distribution of Zn atoms on the surface of this porous material is not homogeneous, and many Zn atoms are immersed in the carbonaceous network, as reported by Díaz Durán and Roncaroli for cobalt [51], hence the lower mass percentage of the FAMES components of the biodiesel (BIOGly@ZIF-8-PCL) synthesized from Gly@ZIF-8-PCL. These structural and textural changes can be evidenced in the N_2_ adsorption–desorption isotherms at 77 K of Gly@ZIF-8 and the pore size distribution curves (Figure 1).

The structural changes suffered in the thermal modification can also be confirmed by comparing the Raman spectra of ZIF-8 with Gly@ZIF-8 (Figure 4A,B). The spectrum of Gly@ZIF-8 did not retain any peaks of ZIF-8, and is characterized by the presence of two high-intensity peaks at 1358 and 1593 cm^−1^ that are attributed to vibrations generated by the formation of new C-C and C-Zn bonds, which were formed by the effect of calcination at 900 °C. The thermal change causes the destruction of the ZIF-8 structure and the structural evolution of a porous carbonaceous network with incorporated zinc atoms.

This carbonaceous network with embedded Zn atoms corresponds to Gly@ZIF-8, which maintains the morphology of ZIF-8, as evidenced by the morphological comparison of these two materials (see Figure 12A,B), hence the importance of using ZIF-8 as a template for the development of new materials, considering that MOF derivatives have similar structural and morphological characteristics to the starting material but with the advantage that the properties of their precursors are amplified in the new materials [64]. In this scenario, the importance of using ZIF-8 as a starting material for the catalytic supports ZIF-8 modified with ethylene glycol (ZIF-8-PGly) and Gly@ZIF-8 resulting from the calcination at 900 °C of ZIF-8-PGly is that ZIF-8 behaves as a structure director that allows the new supports GlyZIF-8 and ZIF-8-PGly to inherit and amplify the structural and morphological characteristics of the parent material (see Figure 12A,B) at the expense of sacrificing textural properties, as shown by the N_2_ isotherms at 77 K and pore distribution curves of the Gly@ZIF-8 (see Figure 1).

Although the supports generated from the modification of ZIF-8 preserved the morphology of ZIF-8 due to the template effect [65], in the coupling with lipase the same effect was not manifested; on the contrary, the interaction in the two cases occurred differently. For the support produced by impregnating ethylene glycol, ZIF-8 catalyzes the self-polymerization of ethylene glycol, which will be responsible for assembling both ZIF-8 and *Pseudomonas Cepacia Lipase* by the hydroxyl groups located at the ends of the polymer (see Figure 10). In the case of the support obtained by calcination, the interaction between the lipase and the substrate occurred because the surface Zn ions attract the hydrophobic amino acids of the *Pseudomonas Cepacia Lipase*.

In relation to the BIOZIF-8-PCL and BIOGly@ZIF-8-PCL biomaterials, despite the fact that both present the *Pseudomonas Cepacia Lipase* in their structure, they do not present compositional and structural similarities (Figure 12A–D). These well-marked differences are due to the fact that the *Pseudomonas Cepacia Lipase* interacts with different substrates, which leads to the formation of different bonds, different geometry, and different distribution of the lipase on the surface on the substrate or support. However, despite the biocatalysts being different, they are effective for the transesterification reaction of acylglycerides present in commercial palm oil, a fact that can be corroborated by comparing the FTIR spectra (Figure 7) and NMR (Figure 8B,C) of the biodiesels BIOZIF-8-PCL and BIOGly@ZIF-8-PCL. The obtained biodiesels have the same composition, with the most abundant FAME being methyl oleate. The biodiesel synthesized with ZIF-8-PLC showed higher mass percentages of FAMEs (see Table 2), which shows that this catalyst has higher performance in relation to Gly@ZIF-8-PLC, as was already shown. This behavior is related to the type of interaction that occurs between the lipase with the substrate: the hydrophobic amino acids present in the *Pseudomonas Cepacia Lipase* interact more easily with groups with high electronic density, such as hydroxyl groups. This greater interaction is proportional to the immobilization of the lipase in the supports used, a fact that can be corroborated and quantified indirectly by comparing the mass percentages of each FAME in the biodiesel produced. Likewise, no signs of substances such as glycerin or oil were found, so it can be inferred that a successful separation was carried out, and in qualitative terms it is inferred that the conversion of acylglycerides present in palm oil to biodiesel components was carried out by complete transesterification, or a mixture of reactions was carried out. Such reactions are common for lipase anchored either on a support derived by impregnation with ethylene glycol or by the support calcined at 900 °C, and they are carried out in two steps. In the first step, the triglycerides present in the oil and the alcohol are converted by the action of the biocatalyst into diglycerides plus fatty acid methyl ester (FAME) [66]. In the second step, the substrate is the diglycerides that react with methanol in excess by the intervention of the biocatalyst and they become monoglycerides and FAME. The monoglycerides produced are soluble in the FAMEs, thus constituting a green diesel (see Figure 13) [67].

Figure 12A,B show that the form of hexapods [68] is present both in ZIF-8 crystals and in Gly@ZIF-8, a fact that corroborates that the synthesized supports maintain the structural characteristics of the ZIF-8 template that originated them [69]. Figure 12C,D clearly show that a successful immobilization of the *Pseudomonas Cepacia Lipase* was carried out on the surface of the support; additionally, it was evidenced that the anchoring of the lipase did not alter the morphological characteristics of the supports.

## 4. Methods and Materials

### 4.1. Reagents Used

ZIF-8 (Sigma-Aldrich, Saint Louis, MO. USA), lipase from *Pseudomonas Cepacia* (Sigma-Aldrich (62309), Saint Louis, MO, USA) methyl alcohol (R.A. Carlo Erba^TM^, Milano, Italy), ethylene glycol (R.A. Carlo Erba^TM^, Milano, Italy), KOH (R.A, brand Carlo Erba™, Milano, Italy), solution 0.1 M CH_3_COOH, buffer at pH 7, commercial biodiesel BIOC, and African palm oil (PO) (certified by the National Federation of Oil Growers, FedePalma, Colombia) were used in this study.

### 4.2. Thermal Modification of the ZIF-8

For the ZIF-8 thermal modification, 1.0022 g was taken and 12.5 mL of ethylene glycol was added. The resulting mixture was left to interact for 13 days. After this time, the ZIF-8 impregnated with ethylene glycol separated from the solution and led to calcination, and the new material was assigned the name of Gly@ZIF8. The calcination conditions to obtain Gly@ZIF-8 were the following: Ramp I: initial temperature 40 °C, final temperature 900 °C for one hour, with heating rate 7 °C.min^−1^.

### 4.3. Isotherms of N_2_ at 77 K

Gly@ZIF-8 was degassed prior to textural characterization. The working conditions for degassing were the following: Station 1: T_1_ = 110 °C, time = 1 h, and Station 2: T_2_ = 250 °C, time = 9.5 h, degassing pressure: 10^−5^ millibars. The textural characterization of Gly@ZIF-8 was performed by physical adsorption of N_2_ at 77 K using a Quantachrome IQ2 sortometer. The results of the gas–solid isotherms were analyzed in different pressure ranges, using the BET method, in which the calculation of the specific surface (P/P^o^ range) was determined using the method proposed by Rouquerol et al. [70]. The density functional theory (range P/P^o^ 10^−7^ − 1) considering different pore models and the effects of surface roughness and heterogeneity (NLDFT and QSDFT) were achieved using the ASQiWin software [71].

### 4.4. Pseudomonas Cepacia Lipase Immobilization

Here, 0.0711 g of ZIF-8 and 0.839 g of Gly@ZIF-8 were taken separately as solutes to prepare 2 solutions with 0.048 and 0.0413 g of *Pseudomonas Cepacia Lipase*. Both solutions were prepared using a buffer at pH 7. The solutions were left to cool for 24 h at a temperature of −80 °C, and then lyophilized.

The temperature-programmed desorption (TPD) was performed for the ZIF-8, Gly@ZIF-8, and Gly@ZIF-8-PCL, using an Auto Chem II 2920 equipped with a thermal conductivity detector (TCD) (Micromeritics, USA). NH_3_ and CO_2_ were used as probe gas for temperature-programmed desorption tests for acidity (NH_3_-TPD) and basicity (CO_2_-TPD) analysis, respectively. The samples were pretreated at 373 K under carrier helium (He) gas flow for 2 h to remove adsorbed water molecules, and then cooled down to the adsorption temperature at 313 K for 1 h. Ammonia (NH_3_) and carbon dioxide (CO_2_) (in 15% NH_3_ and 10% CO_2_ with He balance) used as probe gases were introduced to the sample by continuous flow until saturation, until a stable temperature conductivity detector (TCD) signal was obtained. The physically adsorbed gas on the samples was removed by purging with He flow at 313 K for 30 min. The temperature of the samples was increased to 573 K at a heating rate of 283 K.min^−1^, and the desorbed NH_3_ or CO_2_ gas was detected using a TCD.

### 4.5. Derivatization of Palm Oil for Mass Analysis

The protocol described below is in the process of being published in the journal *Chemosphere*, which shows the derivatization pathways authored by Prof. Dr. Chiara Carazzone and Dr. Gerson Lopez (Department of Chemistry, Universidad de los Andes, Colombia). The derivatization method with BF_3_:MeOH (approximately 10%) and sodium methoxide was performed with 300 µL of a methanolic solution of the sample at a 500 ppm concentration, containing methyl non-nanoate at 300 ppm as an internal standard (IS). Then, 100 µL of sodium methoxide was added to the above solution and heated at 50 °C in an oil bath for 10 min. Subsequently, 180 µL of BF_3_:MeOH solution (approximately 10%) was added and it was heated again in an oil bath at 50 °C for an additional 16 mins. For the extraction, they were added in the following order: 200 µL of water and 200 µL of GC-grade hexane. It was vortexed for 30 s and centrifuged at 10.500 rpm for 5 min, collecting the upper organic phase, transferring to a 2 mL vial with a 300 µL insert, and injecting it into the GC-MS for FAMEs analysis. To prepare sodium methoxide, 574.8 mg of Na was taken and reacted with 50 mL of methanol. Then, the method described above was applied, leaving the derivatizations ready for analysis. 

### 4.6. Transesterification Reaction: Lipase from Pseudomonas Cepacia Supported

Here, 0.1098 g of ZIF-8-PCL and 0.084 g of Gly@ZIF-8-PCL were taken separately, and 1 mL of palm oil and 4 mL of ethanol were added to each reactor and left to react for 24 h. After this time, 1 mL of ethanol was added to each reactor, and it was left to react again for 24 h. Then, the biodiesel was separated from the other phases. The separation of the immobilized lipase was carried out by decantation and subsequent washing with a phosphate buffer at pH 7, followed by lyophilization.

### 4.7. Fourier Transform Infrared Spectrophotometry (FTIR)

The FTIR absorbance spectrum of Gly@ZIF-8 was obtained using the diffuse reflectance technique, with analysis performed on an IRTracer-100 Shimadzu spectrometer in the wavenumber range of 4000–400 cm^−1^. The solid samples were mixed with KBr in a ratio of approximately 1/300, and then the mixture was ground in an agate mortar to a very fine powder. The FTIR absorbance spectrum of Gly@ZIF-8 was obtained using the KBr technique. After drying at 100 °C for 12 h in a vacuum oven, about 300 mg of the fine powder was used to make a pellet. After preparation, the pellet was immediately analyzed, and the spectra were recorded by a series of scans with a resolution of 4 cm^−1^. A pellet prepared with an equivalent amount of pure KBr powder was used as the background. FTIR absorbance spectra for ZIF-8, ZIF-8-PCL, palm oil, BIOC, BIOZIF-8-PCL, and BIOGly@ZIF-8-PCL were obtained by the attenuated total reflectance technique (ATR).

### 4.8. Gas Chromatography Coupled to Mass Spectrometry

To separate, analyze, and quantify the acyglycerides present in palm oil and the fatty acid methyl ester components in the obtained biodiesel, a GC System gas chromatograph was used (Hewlett Packard, coupled with Mass Selective Detector Agilent Technologies 5973 Network Mass Spectrometer). Solutions were prepared at 10.500 ppm of each biodiesel in methanol, then each solution was brought to 1000 ppm, taking 0.020 μL of the mother solutions and adding 980 μL chloroform. Separation was performed on a DB-5MS capillary column (30 m × 0.32 mm, 0.25 μm-thick). The carrier gas was helium with a flow rate of 1.5 mL/min. The column temperature was programmed from 120 to 300 °C at a rate of 10 °C/min. The temperature of both the injector and the detector was set at 250 °C. A sample volume of 1 and 1 μL of dichloromethane as a blank were injected, using a split mode, with a split ratio of 1:10. The mass spectrometer was configured to scan in the range of *m*/*z* 40.00–300.00 with electron impact ionization (El) mode.

### 4.9. Nuclear Magnetic Resonance (NMR)

NMR analyses for commercial biodiesel and synthesized biodiesel were performed using a Bruker AscendTM-400 spectrometer, which operates at 400.13 MHz, RMN 1H, and 13C NMR. Deuterated chloroform (CDCI_3_) and TMS were used as the solvent and internal standard, respectively. The 1H (300 MHz) spectra were recorded with a pulse duration of 30°, a recycle delay of 1.0 s, and 8 scans. The 13C (75 MHz) spectra were recorded with a pulse duration of 30°, a recycle delay of 1.89 s, and 160 scans.

## 5. Conclusions

The synthesis process of the biocatalysts followed two routes. In the first one, the wet impregnation of ZIF-8 with ethylene glycol led to the formation of polyethylene glycol, a polymer that interacts with the imidazole groups of ZIF-8 through the formation of hydrogen bridges, forming a polymeric support intermediate that acts as a support for the immobilization of *Pseudomonas Cepacia Lipase*. The polyethylene glycol anchored to the MOF structure formed rows or layers of hydroxyl groups that facilitated the immobilization of *Pseudomonas Cepacia Lipase* due to the preference of hydrophobic amino acids for groups with a high electron density. This route of synthesis yielded the biocatalyst ZIF-8-PCL as a final product. The second route occurred in a similar way to the first one, but with the difference that once the polymeric intermediate was formed, it was subjected to calcination, with the consequent formation of a carbonaceous network in which the Zn atoms were immersed. However, although the impregnation process sought to protect the structure of ZIF-8, it was not possible to improve its thermal stability. This synthesis route led to the formation of the biocatalyst Gly@ZIF-8-PCL.

In this research, biodiesels BIOZIF-8-PCL and BIO-Gly@ZIF-8-PCL were synthesized from the immobilization of *Pseudomonas Cepacia Lipase* on BIOZIF-8-PCL and BIO-Gly@ZIF-8-PCL from the thermal modification of ZIF-8 impregnated with ethylene glycol, using African palm oil as a raw material. The results showed that the BIOZIF-8- PCL biocatalyst successfully converted the acylglycerides present in palm oil into biodiesel, while the Gly@MOF-199-PCL biocatalyst demonstrated a smaller percentage of transesterification of the acylglycerides present in palm oil, a fact demonstrated by the presence of traces of glycerin in biodiesels BIOZIF-8-PCL and BIOGly@ZIF-8-PCL. The new biocatalysts BIOZIF-8-PCL and BIOGly@ZIF-8-PCL were shown to be effective in the conversion of acylglycerides into biodiesel components.

## Figures and Tables

**Figure 1 molecules-27-05396-f001:**
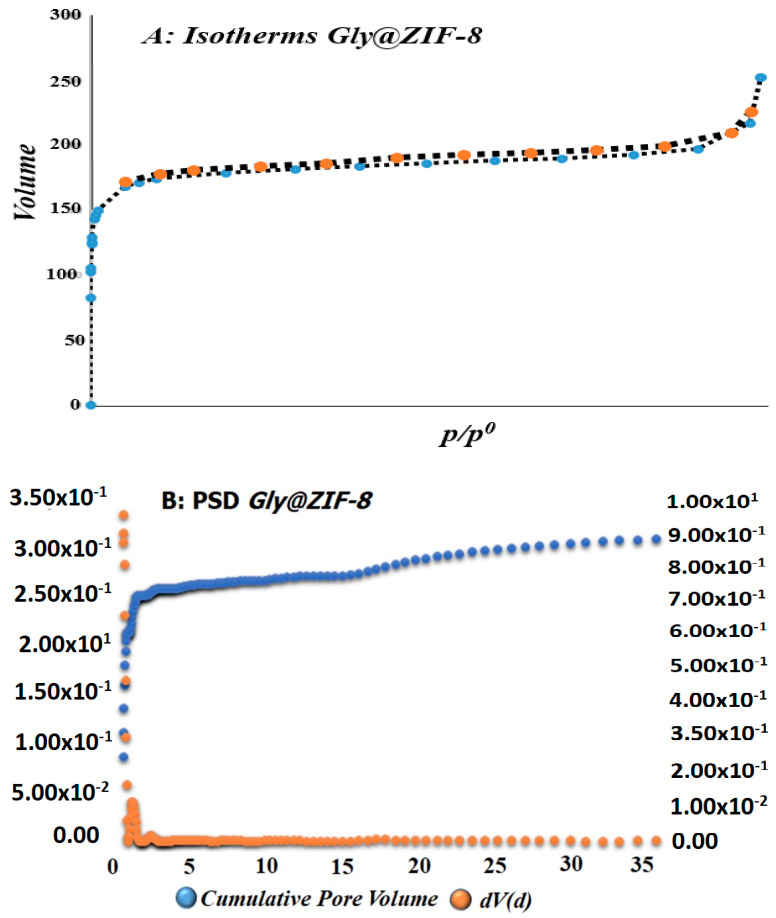
(**A**) Adsorption–desorption isotherms of N_2_ at 77 K of Gly@ZIF-8. (**B**) Pore size distribution curves.

**Figure 2 molecules-27-05396-f002:**
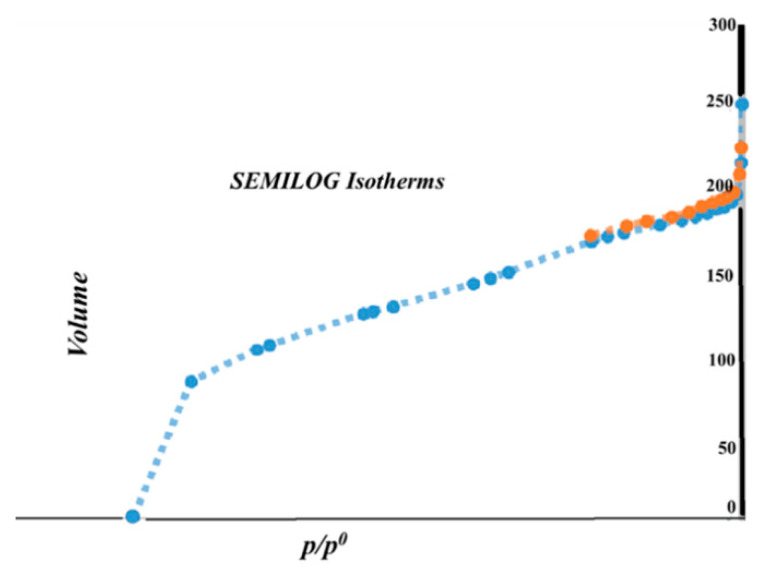
The adsorption–desorption isotherms of N_2_ at 77 K of Gly@ZIF-8.

**Figure 3 molecules-27-05396-f003:**
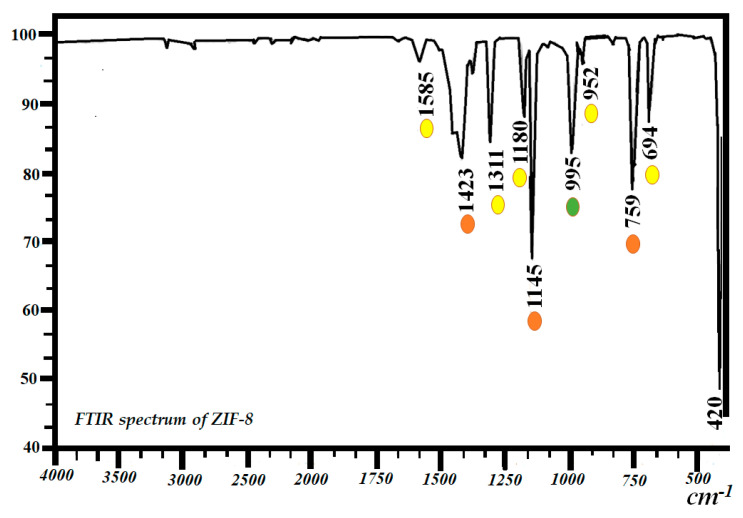
FTIR spectrum of ZIF-8.

**Figure 4 molecules-27-05396-f004:**
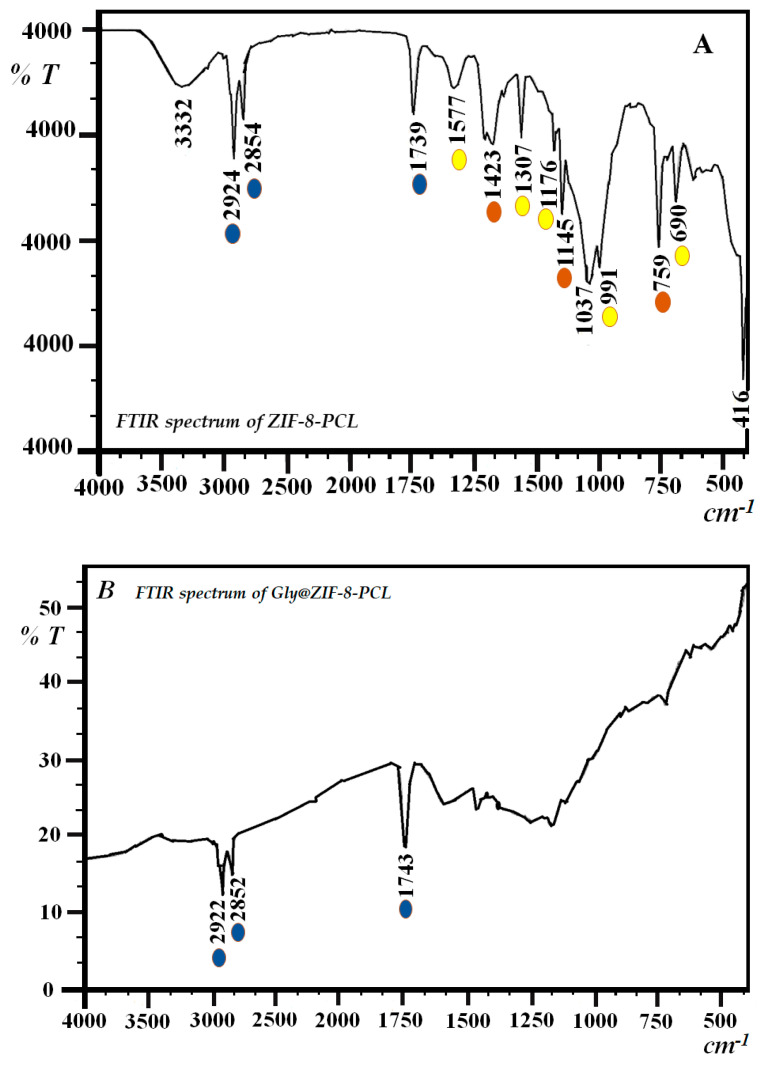
FTIR spectrum of: (**A**) ZIF-8-PCL and (**B**) Gly@ZIF-8-PCL.

**Figure 5 molecules-27-05396-f005:**
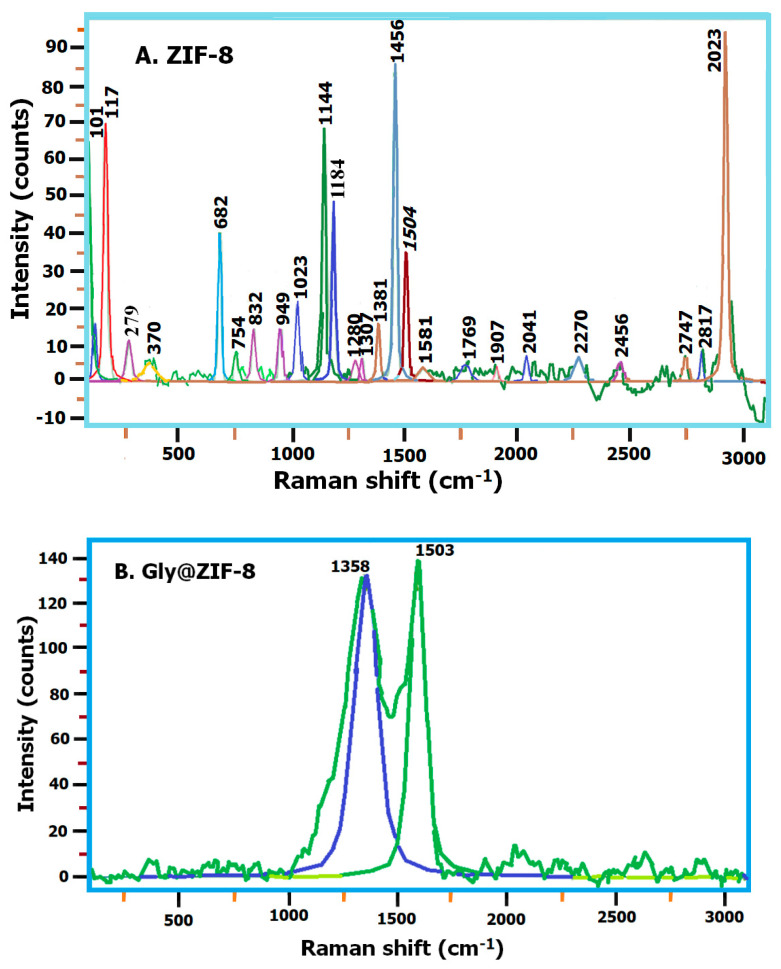
Raman spectrum: (**A**) ZIF-8, (**B**) Gly@ZIF-8, (**C**) ZIF-8-PCL, (**D**) Gly@ZIF-8-PCL, and (**E**) NH_3_ adsorption profiles of the material synthesized in this work. (**F**) CO_2_ adsorption profiles of the material synthesized in this work.

**Figure 6 molecules-27-05396-f006:**
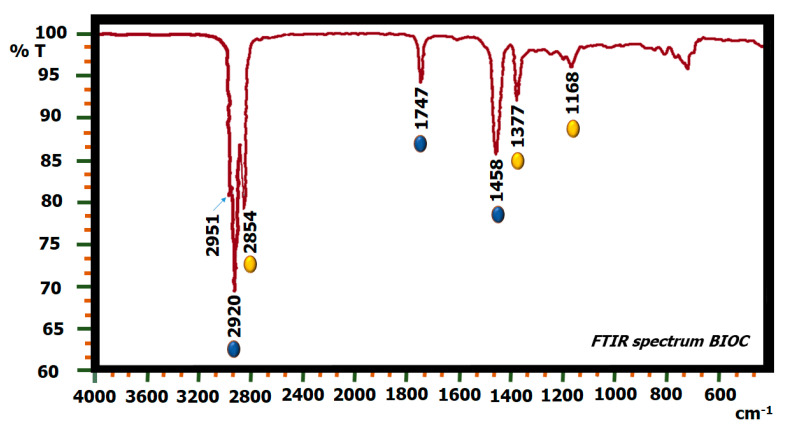
FTIR spectrum of BIOC.

**Figure 7 molecules-27-05396-f007:**
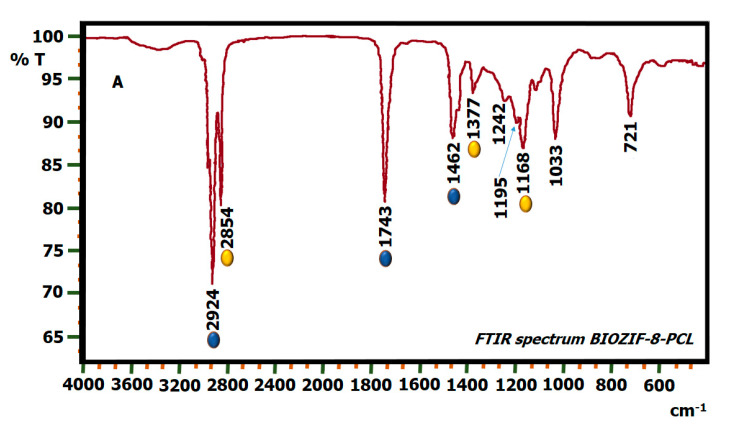
FTIR spectrum of: (**A**) BIOZIF-8 and (**B**) BIOGly@ZIF-8.

**Figure 8 molecules-27-05396-f008:**
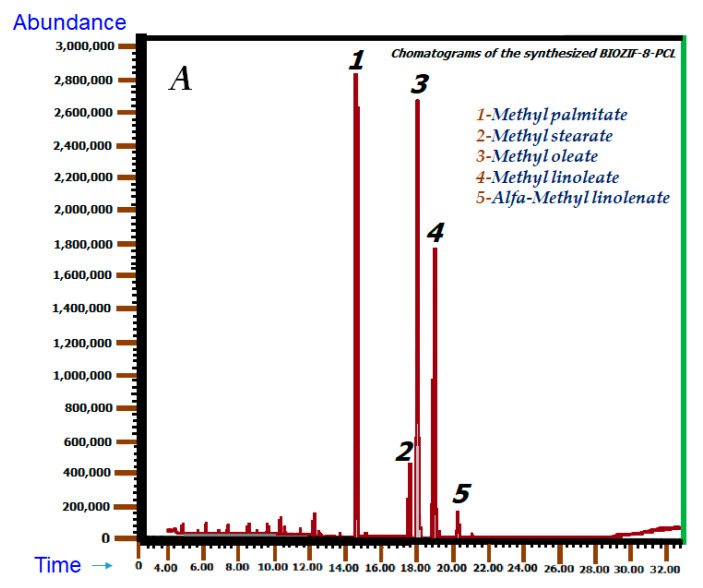
Chromatograms of the synthesized biodiesel: (**A**) BIOZIF-8-PCL and (**B**) BIOGly@ZIF-8-PCL.

**Figure 9 molecules-27-05396-f009:**
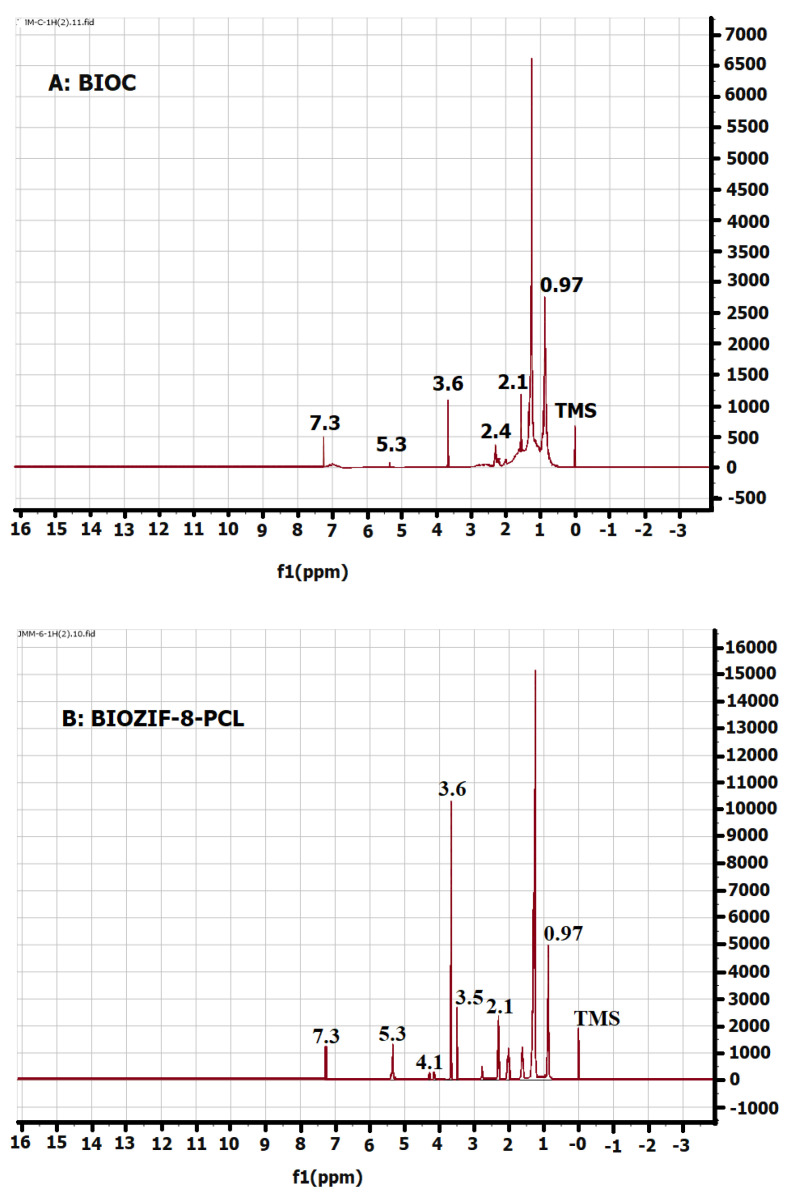
1H NMR spectra of: (**A**) BIOC, (**B**) BIOZIF-8-PCL, (**C**) BIO-Gly@ZIF-8-PCL, and (**D**) palm oil.

**Figure 10 molecules-27-05396-f010:**
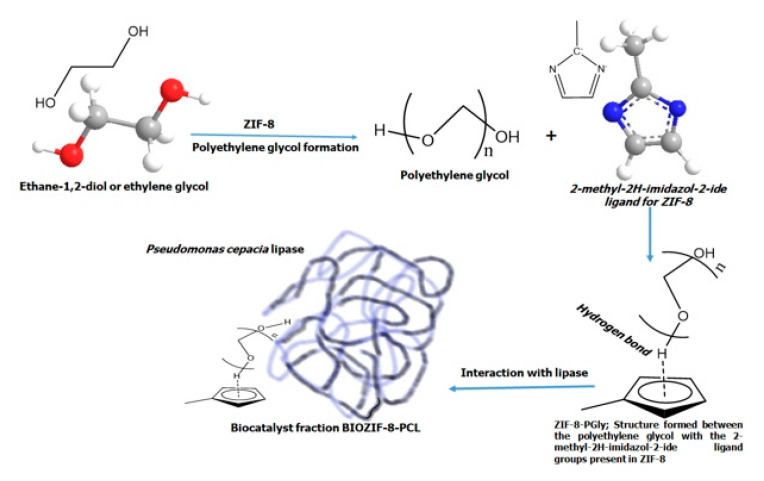
Formation pathway of the ZIF-8-PCL biocatalyst, starting from the interaction of ZIF-8 with ethylene glycol and the formation of polyethylene glycol and ZIF-8-PGly as an intermediate.

**Figure 11 molecules-27-05396-f011:**
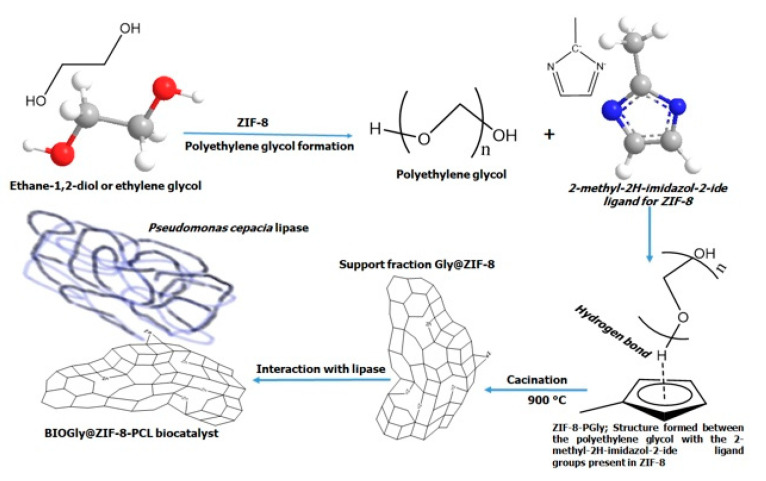
Route of formation of the biocatalyst BIOZIF-8-PCL, starting from the interaction of ZIF-8 with ethylene glycol and the formation of polyethylene glycol and ZIF-8-PGly as an intermediate.

**Figure 12 molecules-27-05396-f012:**
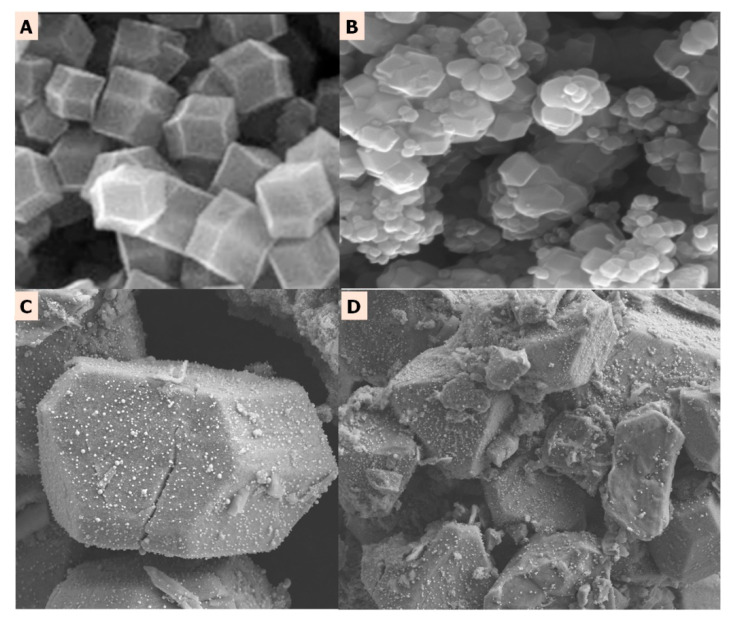
SEM images of (**A**) ZIF-8 surface. (**B**) Gly@ZIF-8 surface, (**C**) ZIF-8-PCL surface, and (**D**) Gly@ZIF-8-PCL surface.

**Figure 13 molecules-27-05396-f013:**
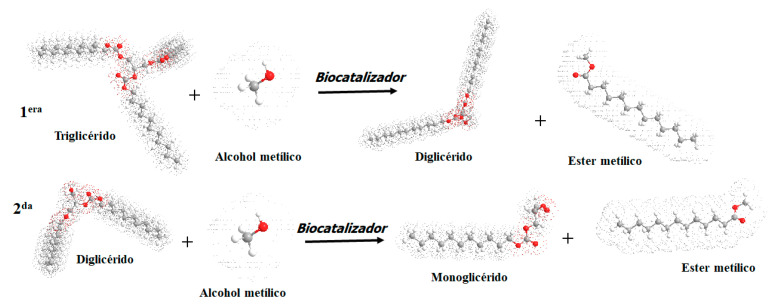
Transesterification reaction. First stage: conversion of triglycerides to diglycerides plus FAME. Second stage: conversion of diglycerides to monoglycerides plus FAME (source: made with ChembioDraw 2D professional18.1 Software. PerkinElmer).

**Table 1 molecules-27-05396-t001:** Textural properties and acid and base amounts of the prepared biocatalysts.

Sample	SBET ^a^ (m^2^g^−1^)	Vpore ^b^ (cm^3^∙g^−1^)	Pore Size ^c^ (nm)	Acid ^d^ Amount (mmol∙g^−1^)	Base ^e^ (mmol∙g^−1^)
ZIF-8	1733	0.609	0.64	0.24	0.08
Gly@ZIF-8	702	0.306	0.69	0.35	0.13
Gly@ZIF-8-PCL	654	0.424	0.53	0.42	0.17

^a^ Specific surface area determined by the BET method. ^b^ Total pore volume. ^c^ Calculated from micropore (MP). ^d^ Obtained from NH3-TPD. ^e^ Obtained from CO2-TPD.

**Table 2 molecules-27-05396-t002:** CG-MS chemical composition of the comparative biodiesels and the synthesized biodiesels.

Asigne Compound	BIOZIF-8-PCL			BIOGly@ZIF-8-PCL			Molecular Formula
	Retention Time (min)	Area	% Mass	Retention Time (min)	Area	% Mass	
Methyl palmitate (1)	14.448	106,823,904	29.12	14.504	263,962,120	24.24	C₁₇H₃₄O₂
Methyl stearate (2)	17.471	18,268,908	4.98	17.556	57,724,642	5.30	C_19_H_38_O_2_
Methyl oleate (3)	17.903	136,362,997	37.17	18.007	372,716,433	34.23	C₁₉H₃₆O₂
Methyl linoleate (4)	18.862	85,866,896	23.41	18.954	283,399,032	26.03	C_19_H_34_O_2_
Alfa-methyl linolenate (5)	20.156	7,094,967	1.93	20.200	31,862,269	2.93	C_19_H_32_O_2_

## Data Availability

Not applicable.

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
