# Peer review of "Biocatalysts Synthesized with Lipase from Pseudomonas cepacia on Glycol-Modified ZIF-8: Characterization and Utilization in the Synthesis of Green Biodiesel"

_molecules, 2022, doi:10.3390/molecules27175396_

Round 1

Reviewer 1 Report

The article can be accepted after the following revisions:

1, the acidity of the used catalysts should be added including NH3-TPD, and Py-IR. Moreover, the correationship between the acidity, BET datas, and the catalytic properties should be discussed in the revision.

2. the reaction was conducted in a liquid-solid system. however, all the characterization was made in a gas-solid system. It is suggested that the relationship between the two systmes should be discussed.

3. In the introduction parts, the content of this article should be depicted in a seperate paragraph.

Author Response

Bogota DC August 16, 2022.

Reference: Letter responses to reviewer 1, Journal Molecules.
Manuscript ID: molecules-1860492
In my capacity as corresponding author in the article entitled “Biocatalysts synthesized with Lipase from Pseudomonas Cepacia on Glycol-modified ZIF-8: Characterization and Utilization in the Synthesis of Green Biodiesel”, I allow myself to send the response letter to each of the suggestions made by reviewer 1, which, as can be seen in the paper, have been accepted and included.

1. The acidity of the catalysts used, including NH3-TPD, and Py-IR, should be added. Furthermore, the correlation between acidity, BET data and catalytic properties should be discussed in the review.
Answer: Regarding the reviewer's suggestion to include Py-IR spectroscopy results, we agree with him in the sense that these show the acid sites of catalysts with pyridine as probe molecule.
However, we have included the NH3-TPD and CO2-TPD tests, which show the acidity and baseness of the catalysts, so we do not include the Py-IR tests, which we did not determine for these catalysts.

2. The reaction was carried out in a liquid-solid system. However, all characterization was performed in a gas-solid system. It is suggested that the relationship between both systems be discussed.

Answer: The supports, biocatalysts and biodiesel were synthesized using a liquid-solid system. The first products of this process, that is, the supports, were characterized in a gas-solid system to determine and compare the surface area and pore size, an analysis of great importance to verify the change in textural properties, this change is necessary due to that it is necessary to increase the pore size, which will allow or facilitate the anchoring of Pseudomonas cepacia lipase.

3. In the parts of the introduction, the content of this article must be represented in a separate paragraph.

Answer: A paragraph was added at the end of the introduction, which describes the content of this article.

Thanks to the correct suggestions of reviewer 1, a clearer paper has now been achieved and where the novelty of the research is show

Juan Carlos Moreno Pirajan, PhD
Full Professor
University of the Andes, Colombia

Reviewer 2 Report

This paper shows the enzymatic production of biodiesel by using Pseudomonas cepacia lipase on Glycol-modified ZIF-8. This study is of interest. The methodology used throughout the experimental work was suitable. However, some revisions are required before it could be considered for publication as follows:
1. In the Introduction section, the background on the lipase immobilization is poor and superficial. The literature review on the latest development in the binding of lipase needs to be substantially enhanced. Different kinds of supports have already been exploited for the lipase immobilization, thus their immobilization tactics should be discussed in this section. The recently published articles are recommended to be included, for example: Food Chemistry, 2018,257:15-22; Energy & Fuel 2014;28:2624-2631; Renewable Energy, 2022, 2022,185:1362-1375; Food Chemistry, 2017; 227:397-403; Fuel, 2021; 291:120126
). More knowledge in the literature on this aspect is recommended to be discussed.

2. The more detailed transesterification process should be presented in your corrected version. The oil conversions or biodiesel yields should be reported in your paper. And, the method to determine the biodiesel yield should also be provided.

3. Elaborate further about the lipase immobilization mechanism in terms of stability for the proposed immobilization methodology.

4. Section 2.4 and 2.5 are wordy and little interest, thus they need to be shortened.

5. The reusability of the immobilized lipase is of great importance especially in the large-scale applications. Is the immobilized lipase reusable?? How is it separated from the reaction mixture after the reaction?

6. The reference needs to be optimized.

Major revisions are necessary to the current manuscript for re-submitting for the publication in the journal.

Author Response

Bogota DC August 16, 2022.

Reference: Letter responses to reviewer 2, Journal Molecules.
Manuscript ID: molecules-1860492

A greeting.
As corresponding author of the article entitled "Biocatalysts synthesized with Lipase from Pseudomonas Cepacia on Glycol-modified ZIF-8: Characterization and Utilization in the Synthesis of Green Biodiesel", I enclose in this letter the respective responses to reviewer 2, with their due explanations. The suggestions of this distinguished reviewer were carefully applied

1. In the Introduction section, the background on lipase immobilization is poor and sketchy. The literature review on the latest advances in lipase immobilization needs to be substantially improved. Different types of supports for lipase immobilization have already been exploited, so their immobilization tactics should be discussed in this section. It is recommended to include recently published articles, for example: Food Chemistry, 2018,257:15-22; Energy & Fuel 2014;28:2624-2631; Renewable Energy, 2022, 2022, 185:1362-1375; Food Chemistry, 2017; 227:397-403; Fuel, 2021; 291:120126). A greater knowledge in the literature on this aspect is recommended to be discussed.

Answer: In relation to the scarce information on the background on the immobilization of lipases, information concerning this aspect was incorporated into the introduction.

As shown below:

This modification of the textural properties allows the good anchoring of the enzyme in the support. The lipases of Pseudomonas Cepacia have a highly open configuration (32,33), therefore the substrate with which it interacts must have the sites with the appropriate conformation. that facilitates its anchoring. The Pseudomonas cepacia lipase is characterized by presenting thermal resistance and tolerance to various substances, especially alcohols (34), if we add to these properties a good anchoring of this enzyme, its thermal resistance would be increased and therefore the resulting biocatalysts can be used in transesterification reactions that exceed the thermal limits of free lipase (35). There have been many efforts to improve the problems associated with the use of free lipases in transesterification reactions (36), however immobilization is the best mechanism to overcome the drawbacks associated mainly with the deactivation of lipases (37). The immobilization of lipases, in addition to not generating by-products (38) in the process of converting acylglycerides to biodiesel components, has the additional benefit that the catalyst can be recovered (39). Several works have shown that the supports produced by the modification of the modified ZIF-8 constitute excellent templates (40) for the anchoring of enzymes, especially lipases from Pseudomonas Cepacia (41-46). The success of the support interaction derived from ZIF-8 and lipase lies in the fact that the structure of ZIF-8 can assume new configurations in the presence of molecules with high electronic density (47), for this particular case, for a first synthesis route of supports to immobilize lipase from Pseudomonas Cepacia, ethylene glycol was used as an agent for expanding the structure of ZIF-8

2. The more detailed transesterification process must be presented in its corrected version. 3. Oil conversions or biodiesel yields should be reported in your article. And, the method to determine the biodiesel yield must also be provided.

Answer: In the analysis of results it became more extensive and the transesterification process is analyzed and described in much more detail. As shown below

or a mixture of reactions was carried out, such reactions are common for lipase anchored either on a support derived by impregnation with ethylene glycol or by the support calcined at 900 °C, and are carried out in two steps. In the first step, the triglycerides present in oil and alcohol are converted by the action of the biocatalyst into di-glycerides plus fatty acid methyl ester (FAME) (Reference), in the second step the substrate is the diglycerides that react with the excess methanol by the intervention of the biocatalyst and they become monoglycerides and FAME, the monoglycerides produced are soluble in the FAMEs, thus constituting a green diesel (Reference).

Scheme 1. Transesterification reaction. A: first stage, conversion of triglycerides to diglycerides plus FAME. B: Second stage, conversion of diglycerides to monoglycerides plus FAME, (Source: made with ChembioDraw 2D professional18.1 Software. PerkinElmer).

3. Explain in more detail the lipase immobilization mechanism in terms of stability for the proposed immobilization methodology.

Answer: An extension is introduced, which explains the locking mechanism in more detail.

As shown below

Although the supports generated from the modification of ZIF-8 preserve the morphology of ZIF-8 due to the template effect (Guo, X., Geng, S), the coupling with lipase does not show the same effect, all Otherwise, the interaction in the two cases occurs differently. For the support produced by impregnating ethylene glycol, ZIF-8 catalyzes the self-polymerization of ethylene glycol, which will be responsible for assembling both ZIF-8 and Pseudomonas cepacia lipase by the hydroxyl groups located at the ends of the polymer see Figure 10. In the case of the support obtained by calcination, the interaction between the lipase and the substrate occurs because the surface Zn ions attract the hydrophobic amino acids of the Pseudomonas cepacia lipase.

4. Sections 2.4 and 2.5 are long and of little interest, so they should be shortened.

Answer: these two sections describe the results in relation to FTIR and Chromatography techniques. The interpretation of the spectra and schedule is quite short, these sections are long due to the number of graphs, necessary to allow the comparison of the different materials and synthesized biodiesel.

5. The reuse of immobilized lipase is of great importance, especially in large-scale applications. Is immobilized lipase reusable? How is it separated from the reaction mixture after the reaction?

Answer: The separation of the immobilized lipase was carried out by decantation and subsequent washing with a phosphate buffer at pH 7, followed by lyophilization.

6. It is necessary to optimize the reference.

Answer: References were optimized

We thank reviewer 2 for each of their suggestions, which have allowed our paper to have a better level and be much clearer and see its novelty in order to get it published.

Sincerely yours,

Juan Carlos Moreno Pirajan, PhD
Full Professor
University of the Andes

Round 2

Reviewer 2 Report

This paper can be accepted for publishing in this Journal.